# Improved Likelihood Probability in MIMO Systems Using One-Bit ADCs

**DOI:** 10.3390/s23125542

**Published:** 2023-06-13

**Authors:** Tae-Kyoung Kim

**Affiliations:** Department of Electronic Engineering, Gachon University, Seongnam 13120, Republic of Korea; tk415kim@gmail.com

**Keywords:** multi-input multi-output systems, one-bit analog-to-digital converters, maximum likelihood probability, weighted combining

## Abstract

This study considers an improved likelihood probability in multi-input multi-output (MIMO) systems using one-bit analog-to-digital converters (ADCs). MIMO systems using one-bit ADCs are known to exhibit from performance degradation because of inaccurate likelihood probabilities. To overcome this degradation, the proposed method leverages the detected symbols to estimate the true likelihood probability by combining the initial likelihood probability. An optimization problem is formulated to minimize the mean-squared error between the true and combined likelihood probabilities, and a solution is derived using the least-squares method. Simulation results show that the proposed method obtains a signal-to-noise gain of approximately 0.3 dB to achieve a frame error rate of $10^{-1}$ compared to conventional methods. This improvement in performance is attributed to the enhanced reliability of the likelihood probability.

## 1. Introduction

Millimeter-wave (mmWave) systems are key technologies in current wireless applications [1,2,3,4,5,6,7]. A large bandwidth is easily available in mmWave bands, thus enabling data rates in the order of gigabits-per second to be achieved. However, technical challenges exist when exploiting this large bandwidth, such as increased power consumption. Power consumption increases linearly with bandwidth and exponentially with the number of quantization bits [8,9]. Thus, conventional systems that use approximately 12 quantization bits at the receiver are unsuitable for mmWave systems. To overcome this limitation, communication systems using low-resolution analog–digital converters (ADCs) have been introduced [10,11,12,13,14,15]. These systems enable the use of mmWave bands by considerably reducing the power consumption with only slight degradation in system performance.

In recent studies, one-bit ADCs have been introduced, which accelerate the realization of mmWave systems by addressing the power-consumption problem [16,17,18,19,20,21]. However, the MIMO systems using one-bit ADCs suffers from performance degradation. Maximum-likelihood (ML) detection, which was originally introduced in [18], has been considered to achieve optimal performance. However, owing to the non-linearity of one-bit ADCs, the original ML detection exhibits unfavorable complexity at the receiver, thus necessitating a convex optimization solution [19]. Hence, practical algorithms were introduced [20,21]. Sphere decoding is a conventional low-complexity algorithm for ML detection and its application in one-bit ADCs was proposed [20]. Meanwhile, soft decisions for ML detection have been investigated in [21]. However, these methods assume perfect channel information at the receiver, which cannot be achieved in one-bit ADCs due to their non-linear properties.

Despite its difficulty in improving estimation performance, channel estimation in one-bit ADCs has been widely investigated [22,23,24,25,26,27,28]. Most investigations assumed that the quantization error model is Gaussian [24]. Similar to conventional linear approaches, least-squares and minimum mean-squared error (MMSE) estimations can be easily derived based on the quantization error model in one-bit ADCs [25,26]. However, the linear approach exhibits unsatisfactory performance because the quantization error model is inaccurate. Iterative approaches can improve estimation performance. The expectation-maximization [27] and approximate message passing [28] algorithms are representative methods for one-bit ADCs. However, iterative approaches exhibit considerable inherent complexities. Consequently, the accuracy of the likelihood probability is limited owing to complexity.

Recent investigations have focused on improving the performance of one-bit ADCs by directly obtaining the likelihood probability [29,30,31,32,33,34,35]. These studies do not rely on obtaining channel estimates; instead, they learn the likelihood probability using machine-learning algorithms, such as supervised learning and reinforcement learning (RL). In previous studies [30,31], a supervised-learning algorithm was applied to pilot transmission; however, its performance was dependent on the pilot length and it yields unsatisfactory performance when extended to wireless standards. This problem can be mitigated by using detected data symbols in data transmission [32]; however, only correct symbols can improve the likelihood probability because the detected data symbol may contain errors. Hence, a Markov decision process problem was defined [32], and a corresponding RL algorithm was introduced to calculate the optimal policy. However, the computation of the optimal policy involved significant computational complexity, and the number of candidate symbols increased exponentially with the number of modulation orders and transmit antennas. In [33], a low-complexity algorithm for RL was introduced that exploited a reliability-based policy, which significantly reduced the complexity. Although the RL methods in [32,33] significantly improve performance, the complexity associated with computing the optimal policy is not conducive to the receivers. To address this complexity, this study investigates a new method for enhancing likelihood probability without calculating the optimal policy.

This study proposes an improved likelihood probability method for multi-input multi-output (MIMO) systems that use one-bit ADCs. To achieve this, the proposed method utilizes the detected data symbols to calculate the data-based likelihood probability, which is combined with the pilot-based likelihood probability obtained during pilot transmission. An optimization problem is presented herein for the combined process and a computationally efficient algorithm for solving it is described. The key contributions of this study are summarized as follows:An optimization problem is formulated to improve the pilot-based likelihood probability. In particular, a data-based likelihood probability is defined using detected data symbols, and a weighted combination method is proposed between the pilot- and data-based likelihood probabilities. The optimization problem minimizes the expected mean squared error (MSE) between the true and weighted combined likelihood probabilities. Unlike the likelihood probability in [32], the data-based likelihood probability considers all detected data symbols in its calculation, and the weights are unconstrained rather than summed up to one.A computationally efficient method for solving the optimization problem is proposed. The optimization problem is challenging to solve because the non-linear function in the likelihood probability and the expression of the data-based likelihood probability are mathematically intractable. Hence, the method exploits realized samples to obtain weights for the optimization problem. The expectation of optimization problem is replaced with the average of the realized samples. The solution to the optimization problem is derived using the least-squares method. Because the derived solution is obtained offline, its weights are saved in memory.The effectiveness of the proposed method is demonstrated through simulations. The combined weights for the proposed method are updated from the memory at each end of the data block because updating the weights of every data symbol is burdensome to the memory. Based on simulation results, the proposed method indicates improved performance compared with conventional methods without sacrificing complexity. This is because the proposed method improves the reliability of the likelihood probability by exploiting the detected symbols.

**Related works:** This paper focuses on improving the likelihood probability in MIMO systems using one-bit ADCs. The investigation of likelihood probability in one-bit ADCs was initially conducted in [18,20]. Although the likelihood probability offers optimal performance in one-bit ADCs, its performance remains unsatisfactory. To address this performance limitation, the authors [32,33] introduced the concept of data-based likelihood probability. These methods exploit RL algorithms to obtain the optimal policy. However, due to considerable complexity involved in calculating the optimal policy, the methods in [32,33] are still impractical for wireless applications. This study considers a new method to leverage the data-based likelihood probability without the need for complex calculation. To achieve this, an optimization problem is defined that minimizes the expected MSE between the weighted combination of pilot-based and data-based likelihood probabilities and the true likelihood probability. One key advantage of the proposed method is that it does not require weight calculation during the transmission. Thus, the complexity is significantly reduced compared to [32,33]. To improve the comprehensiveness of this study, the related works are summarized in Table 1. The focus of this study is on data detection in one-bit ADCs. Previous studies have addressed the challenges posed by one-bit ADCs by employing complex machine learning-based algorithms, such as supervised learning and reinforcement learning. However, to reduce the inherent complexity associated with machine learning algorithms, this study focuses on improving likelihood probability for low-complexity data detection.

The remainder of this study is organized as follows. Section 1 describes the frame structure and the MIMO receiver using one-bit ADCs. Section 2 presents the estimated likelihood probability using the detected data symbols and an optimization problem that minimizes the MSE between the true- and estimated-likelihood probabilities. Section 3 explains the proposed method for solving the optimization problems. Section 4 presents the simulation results obtained using the proposed method. Finally, the conclusions are presented in Section 5.

## 2. System Model

This section describes the system model used in this study. The transmitter has $N_{t}$ antennas and the receiver has $N_{r}$ antennas. Each transmission frame comprises a pilot block and *D* data blocks. The number of symbols in the pilot block is denoted by $N_{p}$, whereas the number of symbols in the *d*-th data block is denoted by $N_{d}$, where *d* is an element of set {1,2,…,D} (Figure 1).

The information bits are encoded by a channel encoder with code rate $R_{c}$, and mapped to *M*-ary quadrature amplitude modulation (QAM) symbols $\bar{s}\left[n\right]\in C^{N_{t}\times 1}$ where C is a complex space. K denotes the index set of the candidate QAM symbols and is defined as {1,2,…,K}. $\bar{X}$ denotes the set of candidate QAM symbols and is defined as $\bar{X}=\left\{{\bar{s}}_{1},\ldots ,{\bar{s}}_{K}\right\}$. Its size is $\left|K\right|=M^{N_{t}}$, where the modulation order is *M* and |·| is the cardinality of the set. The wireless channel $\bar{H}\in C^{N_{r}\times N_{t}}$ is assumed to exhibit Rayleigh fading, and the channel is invariant during each frame transmission. When the QAM symbol $\bar{s}\left[n\right]\in \bar{X}$ is transmitted to the wireless channel, the received symbol $\bar{r}\left[n\right]\in C^{N_{r}\times 1}$ is expressed as
(1)$$\begin{array}{cc} \bar{r}\left[n\right] & =\bar{H}\bar{s}\left[n\right]+\bar{n}\left[n\right], \end{array}$$
where the additive white Gaussian noise is denoted by $\bar{n}\left[n\right]$, whose distribution follows $\mathrm{CN}(0,\sigma_{n}^{2})$.

For simplicity, the complex value expression in (Equation 1) is converted into a real-value expression. Subsequently, the real vectors and matrices $r\left[n\right]\in R^{2N_{r}\times 1}$, $H\in R^{2N_{r}\times 2N_{t}}$, $s\left[n\right]\in R^{2N_{t}\times 1}$, and $n\left[n\right]\in R^{2N_{r}\times 1}$, where R is the real space, are defined as
(2)r[n]=Re{r¯[n]}Im{r¯[n]},H=Re{H¯}−Im{H¯}Im{H¯}Re{H¯},s[n]=Re{s¯[n]}Im{s¯[n]},n[n]=Re{n¯[n]}Im{n¯[n]}.
where Re(·) and Im(·) are the real and imaginary components of the complex number, respectively. A one-bit operation is performed on the real and imaginary components of the received symbol r[n], which is expressed in [20,32] as
(3)$$\begin{array}{cc} q[n] & =\mathrm{sign}\left(r\left[n\right]\right)=\mathrm{sign}\left(Hs[n]+n[n]\right), \end{array}$$
where Sign(x) is a function whose output is 1 for x≥0 and −1 otherwise.

ML detection provides the optimal performance when the candidate QAM symbols are transmitted equally, which is expressed as
(4)$$\hat{k}\left[n\right]=\underset{k\in K}{\mathrm{argmax}}P\left(q\left[n\right]|s\left[n\right]=s_{k}\right),$$
where P(·) is the probability operation; sk=[Re(s¯k)T,Im(s¯k)T]T is the candidate QAM symbol; and ${\left(\cdot \right)}^{T}$ is the transpose. Subsequently, $s_{k}$ belongs to the real-valued constellation set $X=\{s_{1},\ldots ,s_{K}\}$. Based on [20], the ML in a one-bit ADC is computed as
(5)$$\begin{array}{cc} P(q\left[n\right]|s\left[n\right]=s_{k}) & =\prod_{r\in R}P\left(q_{r}\left[n\right]|s\left[n\right]=s_{k}\right), \end{array}$$
where the probability $P(q_{r}\left[n\right]|s\left[n\right]=s_{k})$ is expressed as
(6)$$\begin{array}{cc} P(q_{r}\left[n\right]|s\left[n\right]=s_{k}) & =\left\{\begin{array}{cc} p_{r,k}, & q_{r}\left[n\right]=+1\\ 1-p_{r,k}, & q_{r}\left[n\right]=-1 \end{array}\right.. \end{array}$$

Here, $q_{r}\left[n\right]$ is the *r*-th row of q[n] and $r\in R=\{1,2,\ldots ,2N_{r}\}$. The probability $p_{r,k}$ in [20] is computed as
(7)$$\begin{array}{cc} p_{r,k} & =\Phi \left(\sqrt{\frac{2}{\sigma_{n}^{2}}}h_{r}^{H}s_{k}\right). \end{array}$$
where $h_{r}^{H}$ is the *r*-th row of H, ${\left(\cdot \right)}^{H}$ is the conjugate transpose, and Φ(·) is the cumulative distribution of the standard normal random variable.

In this study, $p_{r,k}$ denotes the true likelihood probability computed based on the true channel information H. However, the true channel information is unavailable at the receiver. Hence, channel estimation is performed during pilot transmission. By denoting the channel estimate at the receiver $\hat{H}$, the pilot-based likelihood probability $p_{r,k}^{pi}$ is expressed as
(8)$$\begin{array}{cc} p_{r,k}^{pi} & =\Phi \left(\sqrt{\frac{2}{\sigma_{n}^{2}}}{\hat{h}}_{r}^{H}s_{k}\right). \end{array}$$

Using the results of (Equation 5)—(Equation 7), the ML detection is expressed as
(9)$$\begin{array}{cc} \hat{k}\left[n\right] & =\underset{k\in K}{\mathrm{argmax}}\prod_{r:q_{r}\left[n\right]=1}p_{r,k}^{pi}\prod_{r:q_{r}\left[n\right]=-1}\left(1-p_{r,k}^{pi}\right). \end{array}$$

Here, $\hat{s}\left[n\right]$ is defined as the ML estimate, which is re-expressed as $s_{\hat{k}\left[n\right]}$ using $\hat{k}\left[n\right]$. Representative symbols in this study are summarized in Table 2.

## 3. Optimization Problem

In this section, an optimization problem is defined to minimize the MSE between the true and estimated likelihood probabilities. To obtain the estimated likelihood probability, a data-based likelihood probability was first defined and then calculated based on the detected data symbols. Subsequently, the estimated likelihood probability was obtained by linearly combining the pilot- and data-based likelihood probabilities. The goal of the optimization problem is to determine the optimal combination ratio for this linear combination.

### 3.1. Data-Based Likelihood Probability

This subsection presents the derivation of the data-based likelihood probability based on the detected data symbols. The probability $p_{r,k}$ in (Equation 6) is expressed using the following conditional probability formula,
(10)$$\begin{array}{cc} p_{r,k} & =P(q_{r}\left[n\right]=1|s\left[n\right]=s_{k})\\ \; & =\frac{P(q_{r}\left[n\right]=1,s\left[n\right]=s_{k})}{P(s\left[n\right]=s_{k})} \end{array}$$
where the probabilities in (Equation 10) are, respectively, defined as
(11)$$\begin{array}{cc} P(q_{r}\left[n\right]=1,s\left[n\right]=s_{k}) & =\frac{\sum_{m\leq n}{\tilde{q}}_{r}\left[m\right]1\left\{k\left[m\right]=k\right\}}{n},\\ P(s\left[n\right]=s_{k}) & =\frac{\sum_{m\leq n}1\left\{k\left[m\right]=k\right\}}{n}. \end{array}$$

Here, ${\tilde{q}}_{r}\left[n\right]$ is a function whose value is 1 if $q_{r}\left[n\right]=1$ and 0 otherwise. 1{E} is one if event E is true; otherwise, it is 0. The probabilities in (Equation 11) can not be computed at the receiver because the transmitted symbol index k[m] is not available. Therefore, they are approximated using the detected symbol index $\hat{k}\left[m\right]$, which is expressed as
(12)$$\begin{array}{cc} P(q_{r}\left[n\right]=1,s\left[n\right]=s_{k}) & \approx \frac{\sum_{m\leq n}{\tilde{q}}_{r}\left[m\right]1\left\{\hat{k}\left[m\right]=k\right\}}{n},\\ P(s\left[n\right]=s_{k}) & \approx \frac{\sum_{m\leq n}1\left\{\hat{k}\left[m\right]=k\right\}}{n}. \end{array}$$

These approximations are similar to the true probabilities for a sufficiently large *n* [32]. Subsequently, the data-based likelihood probability is obtained as follows:(13)$$\begin{array}{cc} p_{r,k}^{da} & \approx \frac{\sum_{m\leq n}{\tilde{q}}_{r}\left[m\right]1\left\{\hat{k}\left[m\right]=k\right\}}{\sum_{m\leq n}1\left\{\hat{k}\left[m\right]=k\right\}}. \end{array}$$

### 3.2. Problem Definition

In this subsection, the optimization problem is defined to minimize the expectation of MSE between the true and estimated likelihood probabilities. The estimated likelihood probability ${\hat{p}}_{r,k}$ is defined as a weighted linear combination of the pilot- and data-based likelihood probabilities in (Equation 8) and (Equation 12), respectively. This combination is employed because the data-based likelihood probability $p_{r,k}^{da}$ is considered unreliable for a small *n*, whereas the pilot-based likelihood probability is more reliable in this case. In addition, linear combinations have a simple structure and are mathematically tractable. The estimated likelihood probability is expressed as follows,
(14)$$\begin{array}{cc} {\hat{p}}_{r,k}\left(\alpha ,\beta \right) & =\alpha p_{r,k}^{pi}+\beta p_{r,k}^{da}, \end{array}$$
where α,β∈[0,1]. Using the estimated likelihood probability, and the optimization problem is defined as
(15)$$\begin{array}{c} \left(\alpha ,\beta \right)=\underset{\alpha ,\beta \in [0,1]}{\mathrm{argmin}}E\{∥p_{r,k}-{\hat{p}}_{r,k}\left(\alpha ,\beta \right){∥}^{2}\}, \end{array}$$
where E(·) denotes the expectation operation.

Solving the optimization problem in (Equation 15) directly is difficult due to the non-linear function Φ(·). Moreover, the data-based likelihood probability cannot be readily expressed mathematically in a tractable form. Hence, a least-squares method is instead used to obtain the weights of (α,β).

## 4. Proposed Method

In this section, an improved likelihood probability method is proposed to solve the optimization problem shown in (Equation 15). Owing to its mathematical intractability, the optimization problem is difficult to solve. Hence, the realized samples of the likelihood probabilities were leveraged. Using these samples, the combined weights in the optimization problem were trained using the least-squares solution, which resulted in an improved likelihood probability.

### 4.1. Proposed Method

Because the optimized weights cannot be derived easily from the expectation form in (Equation 15), the expectation form is replaced using the realized samples. When *n* realized samples of likelihood probability are available, the optimization problem can be expressed as:(16)$$\begin{array}{cc} \left(\alpha ,\beta \right) & =\underset{\alpha ,\beta \in [0,1]}{\mathrm{argmin}}E\{∥p_{r,k}-{\hat{p}}_{r,k}\left(\alpha ,\beta \right){∥}^{2}\},\\ \; & =\underset{\alpha ,\beta \in [0,1]}{\mathrm{argmin}}\sum_{m\leq n}{∥p_{r,k}\left[m\right]-\alpha p_{r,k}^{pi}\left[m\right]-\beta p_{r,k}^{da}\left[m\right]∥}^{2}, \end{array}$$
where $p_{r,k}^{\cdot }\left[m\right]$ is the *m*-th realized sample of the likelihood probability obtained at the receiver. The optimization problem in (Equation 16) is suboptimal because the weights are obtained from a limited number of realized samples whose values are incorrect. However, as *n* increases, the optimization problem is approximated as that in (Equation 15) because the values of the realized samples converge.

For mathematical tractability, matrix $A_{n}\in R^{n\times 2}$ and vector $B_{n}\in R^{n\times 1}$ are defined from the realized samples as follows:(17)$$\begin{array}{cc} A_{n} & =\left[{\left[p_{r,k}^{pi}\left[1\right],\ldots ,p_{r,k}^{pi}\left[n\right]\right]}^{T},{\left[p_{r,k}^{da}\left[1\right],\ldots ,p_{r,k}^{da}\left[n\right]\right]}^{T}\right],\\ b_{n} & ={\left[p_{r,k}\left[1\right],\ldots ,p_{r,k}\left[n\right]\right]}^{T}, \end{array}$$
where R is a real number.

Using (Equation 17), the optimization problem can be expressed in the matrix form as follows:(18)$$\begin{array}{c} \left(\alpha ,\beta \right)=\underset{\alpha ,\beta \in [0,1]}{\mathrm{argmin}}{∥b_{n}-A_{n}\left[\begin{array}{c} \alpha \\ \beta \end{array}\right]∥}^{2}. \end{array}$$

This optimization problem is well-known least-squares optimization [36]. From the least-squares solution, the optimal value $\left(\alpha^{★},\beta^{★}\right)$ of (Equation 18) can be derived as:(19)$$\begin{array}{cc} \left(\alpha^{★},\beta^{★}\right) & ={\left(A_{n}^{T}A_{n}\right)}^{-1}A_{n}^{T}b_{n}. \end{array}$$

The optimal weights of $\left(\alpha^{★},\beta^{★}\right)$ converge to the optimal weights in (Equation 15) as *n* increases.

### 4.2. Proposed Receiver

The proposed receiver shown in Figure 2 is described in this subsection. Because the optimization problem in (Equation 15) improves the expected likelihood probability, the likelihood probability is updated for every data block *d*. Initially, the proposed receiver obtains the channel estimates during the pilot transmission. The pilot-based likelihood probability $p_{r,k}^{pi}$ is computed using (Equation 8). During data transmission, the receiver detects the data symbol based on the ML criterion expressed in (Equation 20). Using the estimated likelihood probability, the detection method can be expressed as follows:(20)$$\begin{array}{cc} \hat{k}\left[n\right] & =\underset{k\in K}{\mathrm{argmax}}\hat{P}\left(q\left[n\right]|s\left[n\right]=s_{k}\right)\\ \; & =\underset{k\in K}{\mathrm{argmax}}\prod_{r:q_{r}\left[n\right]=1}{\hat{p}}_{r,k}\prod_{r:q_{r}\left[n\right]=-1}\left(1-{\hat{p}}_{r,k}\right). \end{array}$$

The detected data symbol is used to update the data-based likelihood probability $p_{r,k}^{da}$ based on (Equation 13). In addition, the estimated likelihood probability ${\hat{p}}_{r,k}$ is used in the log-likelihood ratio (LLR) calculation, which is expressed as
(21)$$\begin{array}{cc} LLR(b_{i,j}\left[n\right]) & =\log\frac{P(q\left[n\right]|b_{i,j}\left[n\right]=1)}{P(q\left[n\right]|b_{i,j}\left[n\right]=0)}=\log\frac{\sum_{s_{k}\in X_{i,j}^{1}}P\left(q\left[n\right]|s\left[n\right]=s_{k}\right)}{\sum_{s_{k}\in X_{i,j}^{0}}P\left(q\left[n\right]|s\left[n\right]=s_{k}\right)}\\ \; & \overset{\left(a\right)}{=}\log\frac{\sum_{s_{k}\in X_{i,j}^{1}}\hat{P}\left(q\left[n\right]|s\left[n\right]=s_{k}\right)}{\sum_{s_{k}\in X_{i,j}^{0}}\hat{P}\left(q\left[n\right]|s\left[n\right]=s_{k}\right)}=\log\frac{\sum_{s_{k}\in X_{i,j}^{1}}\left(\prod_{r:q_{r}\left[n\right]=1}{\hat{p}}_{r,k}\prod_{r:q_{r}\left[n\right]=-1}\left(1-{\hat{p}}_{r,k}\right)\right)}{\sum_{s_{k}\in X_{i,j}^{0}}\left(\prod_{r:q_{r}\left[n\right]=1}{\hat{p}}_{r,k}\prod_{r:q_{r}\left[n\right]=-1}\left(1-{\hat{p}}_{r,k}\right)\right)}, \end{array}$$
where estimated likelihood probability in (Equation 20) is applied to true likelihood probability (a) because the true likelihood probability is unknown at the receiver. $b_{i,j}\left[n\right]\in \left\{0,1\right\}$ is the *j*-th bit of the *i*-th stream and $X_{i,j}^{b}$ is the set of symbols $s_{k}$ for which $b_{i,j}=b\in \left\{0,1\right\}$. This LLR value is used to obtain the decoded data ${\hat{b}}_{i,j}$ from the channel decoder.

After the data block ends, the proposed receiver updates the estimated likelihood probability ${\hat{p}}_{r,k}$ using the pilot- and data-based likelihood probabilities. The trained weights $\left(\alpha^{★},\beta^{★}\right)$ in memory are used based on the data block number *d*. The estimated likelihood probability was used for the subsequent data block. The sequential algorithm for the proposed receiver is presented in Algorithm 1.
**Algorithm 1:** Proposed Method1 **Pilot transmission**2 Compute initial channel estimate $\hat{H}=\left[{\hat{h}}_{1},\cdots ,{\hat{h}}_{N_{r}}\right].$3 Obtain pilot-based likelihood probability $p_{r,k}^{pi}$, r∈R,k∈K using (Equation 8).4 Set initial estimated likelihood probability ${\hat{p}}_{r,k}=p_{r,k}^{pi}$, r∈R,k∈K.5 **Data transmission**6 **for**
*d=1 to D*
**do**
(7        **for**
*n=1 to $N_{d}$
***do**
(8              Detect data symbol based on ML criterion (Equation 20).9              Calculate data-based likelihood probability $p_{r,k}^{da}$ using (Equation 13).10             Compute LLR based on (Equation 21).11             Obtain decoded data ${\hat{b}}_{i,j}$ from channel decoder.12        **end**(13        Obtain trained weights (α,β) from memory obtained using (Equation 19).14        Update estimated likelihood probability ${\hat{p}}_{r,k}=\alpha p_{r,k}^{pi}+\beta p_{r,k}^{da}$ from (Equation 14).15 **end**(

**Remark (dependent parameters):** The combined weights of the proposed method are trained from the realized samples offline. Thus, the statistical property of the realized samples is important to determine the combining weights (α,β). This statistical property differs from parameters, such as the signal-to-noise ratio (SNR), channel, number of data blocks, number of transmit and receive antennas, modulation order, and coding schemes (see Figure 2). In addition to the parameters above, the parameters that affects the system performance can be input to train the estimated likelihood probability. Thus, the optimal weights (α,β) should be trained by changing the values of the parameters.

**Remark (memory):** Owing to the dependent parameters, many inputs are required for the proposed method. Specifically, the data symbol index *n* significantly increases the memory size. Saving all the parameters every data symbol requires a large memory size, and updating the weights every data symbol increases communication latency. This is because frequent weight updates inevitably lead to increased memory access time. To address this issue, the weights (α,β) are updated only after the end of data block *d*. This update method significantly reduces the memory size and the number of weight update. For example, when $N_{d}=128$ and D=40, a total of 10,280 weights should be saved in the memory, with 10,280 weight updates. However, the proposed method saves only 80 weights in memory, resulting in a corresponding reduction to 80 weight updates. The updated method is described in Algorithm 1.

**Remark (complexity):** To provide a comparison of complexity, Table 3 summarizes the number of real multiplications required to obtain the estimated likelihood probability. For comparison, conventional RL method [32] is introduced. In the conventional RL method, the main complexity arises from calculating the optimal policy, and the corresponding number of real multiplications is listed. In contrast, the proposed method does not require this calculation and only necessitates updating the likelihood probability at the end of each data block. Table 3 presents an example of the complexity for each block when considering $N_{t}=4$, $N_{r}=8$, M=2, $N_{d}=256$, and D=40. In this example, the number of real multiplications required to obtain the estimated likelihood probability is approximately $1.43\times 10^{-4}$ times that of the conventional RL method.

## 5. Simulation Results

The performance of the proposed method was evaluated in terms of the frame error rate (FER) and MSE. The antenna configuration is $\left(N_{t},N_{r}\right)=\left(4,8\right)$ as a practical case of 3GPP specifications. Considering that the maximum number of streams is 8 in 3GPP specifications, $N_{t}=4$ is a practical choice. For the given transmit antennas, $N_{r}=8$ provides a satisfactory FER performance within a practical SNR range in MIMO systems using one-bit ADCs [29,30,31,32,33,34,35]. Each link between the transmit and receive antennas was assumed to be quasi-static Rayleigh fading. The information bits were encoded using a turbo encoder at a rate $R_{c}=\frac{1}{2}$. To improve the performance, a 16-bit cyclic redundancy check was applied, as explained in [32]. A modulation order of four (4−QAM) was applied to the encoded bits, which resulted in a constellation size of $K=M^{N_{t}}=256$. The frame structure comprised one pilot with $N_{p}=32$ and D=40 data blocks with $N_{d}=128$. A linear MMSE method was applied to obtain the pilot-based likelihood probability during the pilot transmission. The SNR is defined as $\frac{1}{\log_{2}\left|X\right|\sigma_{n}^{2}}$.

For performance benchmarking, the following methods were considered:**PCSI:** As an optimal case, the likelihood probability $p_{r,k}$ is obtained based on perfect channel information.**CE:** This is a conventional method in which the likelihood probability $p_{r,k}^{pi}$ is computed based on the channel estimates obtained during the pilot transmission. In this method, the likelihood probability remains unchanged during data transmission.**Optimal:** This method obtains the optimal weights (α,β) in the optimization problem (Equation 15) via exhaustive search, where the resolution of the weights is set to 0.01.**Conv RL:** Weights are obtained using the RL method presented in [32]. In this RL method, the sum of the weights is always 1, i.e., β=1−α. This is expressed as
(22)$$\begin{array}{cc} {\hat{p}}_{r,k} & =\alpha p_{r,k}^{pi}+\left(1-\alpha \right)p_{r,k}^{da}, \end{array}$$
where $\alpha =\frac{1}{\left|R||K\right|}\sum_{r\in R}\sum_{k\in K}\alpha_{r,k}$. Unlike the method presented in [32], the Conv RL method obtains the weights via an offline manner. Subsequently, the obtained weights are applied in the weighted combination method expressed in (Equation 14).

In Figure 3, the combined weights for the proposed method are shown based on *d*. As the number of data blocks increases, the accuracy of the data-based likelihood probability increases. Thus, the weight β for the data-based likelihood probability increases, whereas the weight α for the pilot-based likelihood probability decreases. The sum of weights remains as approximately 1 as the number of data blocks increases. In addition, when the SNR increases, the convergence speed increases because the correctly detected data symbols improve the accuracy of the data-based likelihood probability. Thus, the weight of the data-based likelihood probability has a higher value than that at a lower SNR. The combined weights for the proposed method were simulated based on the SNR, as shown in Figure 4. As shown in this figure, the weight β for the data-based likelihood probability increases with the SNR. Compared with Figure 3, the slope of the weights in Figure 4 increases linearly with the SNR.

In Figure 5, the weights of the conventional RL method are compared with those of the proposed method. Similar to the weights in Figure 3, the weights for the data-based likelihood probability in the conventional RL method increases with *d*. However, the convergence speed of the conventional RL method is higher than that of the proposed method. This is because the weights for the conventional RL method are obtained with respect to the instantaneous MSE not the expected MSE. Thus, despite an SNR of −4 dB, the weights (α,β) almost saturated. The FER and MSE were compared using the above-mentioned weights, as shown in Figure 6 and Figure 7.

The performance of the proposed method depends on the number of antennas used. To verify the effect of the number of antennas, the proposed method was simulated based on the number of transmit antennas $N_{r}$. As shown in Figure 8, the weight β for the data-based likelihood probability was higher when $N_{16}$ was used compared with when $N_{8}$ was used. This is because increasing the number of receive antennas improves the reliability of the detected data symbols. Thus, the accuracy of the data-based likelihood probability increases. Because the optimal weights for the proposed method differs from the system parameters, various values of the weights are saved in memory based on the system environment. Subsequently, based on the system input parameters, the weights are used to the combine pilot- and data-based likelihood probabilities.

The FERs for different detection methods are illustrated in Figure 7 to demonstrate the effectiveness of the proposed method. First, to achieve an FER of $10^{-1}$ at 4×8 antennas, the proposed method exhibited an SNR gain of approximately 0.2 and 0.3 dB compared to conventional RL and CE methods, respectively. In addition, the FER of the proposed method was similar to that of the optimal case. This is attributed to the fact that the proposed method improves the likelihood probability toward the optimal likelihood probability. Second, the FER decreases as the number of receive antennas increases. All the detection methods show an improvement in FER due to the diversity gain associated with a large number of receive antennas. In particular, at 4 antennas, the proposed method showed an SNR gain of approximately 0.2 dB compared to CE method, while achieving a similar FER to the conventional RL. Note that the proposed method can obtain the weights more easily than the conventional RL.

Figure 8 shows the MSE between the true and estimated likelihood probabilities, i.e., $E\{p_{r,k}-{\hat{p}}_{r,k}\}$ for 4×8 MIMO systems. First, the proposed method achieved a lower MSE than the conventional RL. In particular, to achieve an MSE of −18 dB, the proposed method requires approximately 15 number of data blocks, while the conventional RL requires approximately 35 number of data blocks. This indicates that the proposed method achieves more accurate likelihood probability with fewer data blocks. Moreover, the MSE of the proposed method approached that of the optimal case. This is because the proposed method determines weights to minimize the expected MSE between the true and estimated likelihood probabilities. Note that the slope of the optimal case was not smoothed because of the resolution limitation of exhaustive search.

## 6. Conclusions

A novel approach for improving the likelihood probability in MIMO systems that use one-bit ADCs was presented herein. The proposed method leverages detected data symbols to obtain a data-based likelihood probability, which is then combined with the pilot-based likelihood probability using a weighted approach. The optimization problem for the weighted combination was formulated by minimizing the expected MSE, and then a computationally efficient solution to the problem was derived. Simulation results demonstrated that the proposed method achieved an approximate SNR gain of 0.3 dB compared to conventional method by enhancing the accuracy of the likelihood probability.

An interesting direction for future research is to extend to the approach to MIMO systems using low-resolution ADCs, specifically with 2–3 bits of resolution. To achieve this extension, the pilot-based likelihood probability can be obtained using the results from [11]. However, the data-based likelihood probability for the extended system is currently unknown, necessitating further investigation. Another area for future research is to obtain the weights in the optimization problem through an online manner. The proposed weight are currently obtained using off-line manner, which makes it challenging to capture the variations in the channel. To address this limitation, investigating weight update based on the current channel conditions is necessary.

## Figures and Tables

**Figure 1 sensors-23-05542-f001:**
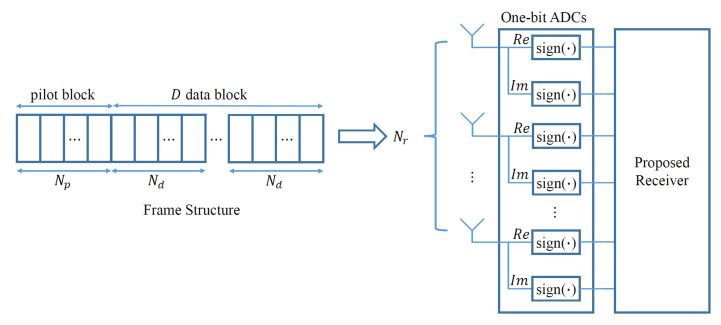
Frame structure and MIMO receiver using one-bit ADCs considered in this study.

**Figure 2 sensors-23-05542-f002:**
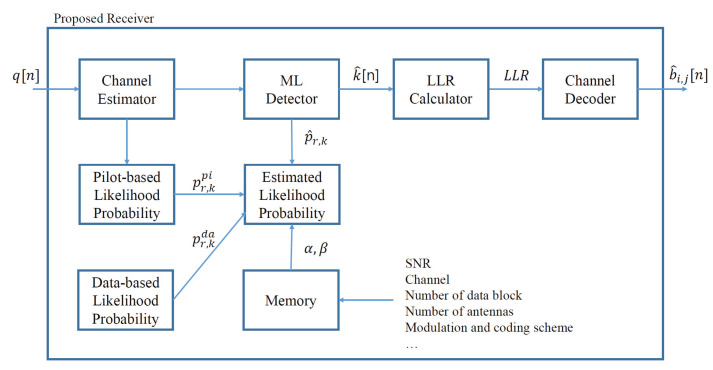
Proposed receiver with the linear combination of pilot- and data-based likelihood probabilities.

**Figure 3 sensors-23-05542-f003:**
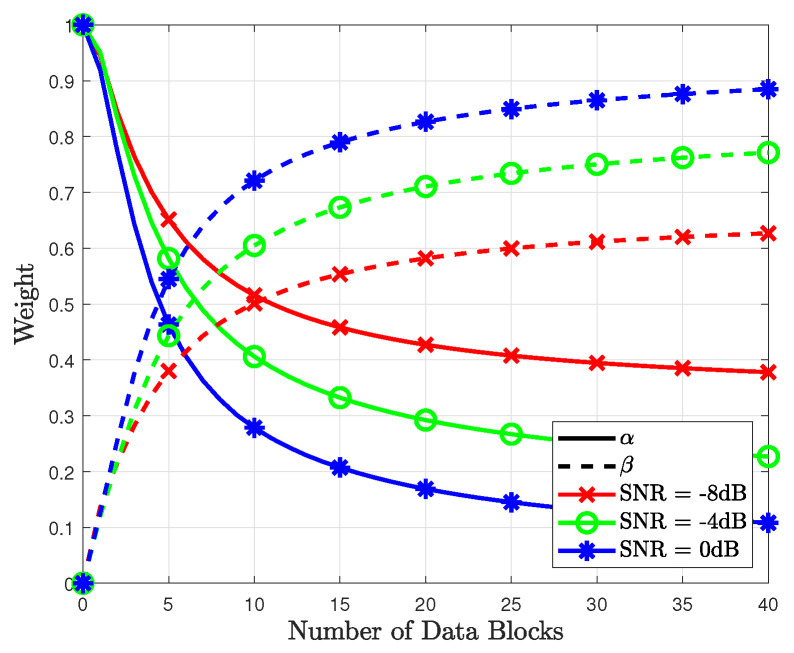
Combined weights for the proposed method based on number of data blocks.

**Figure 4 sensors-23-05542-f004:**
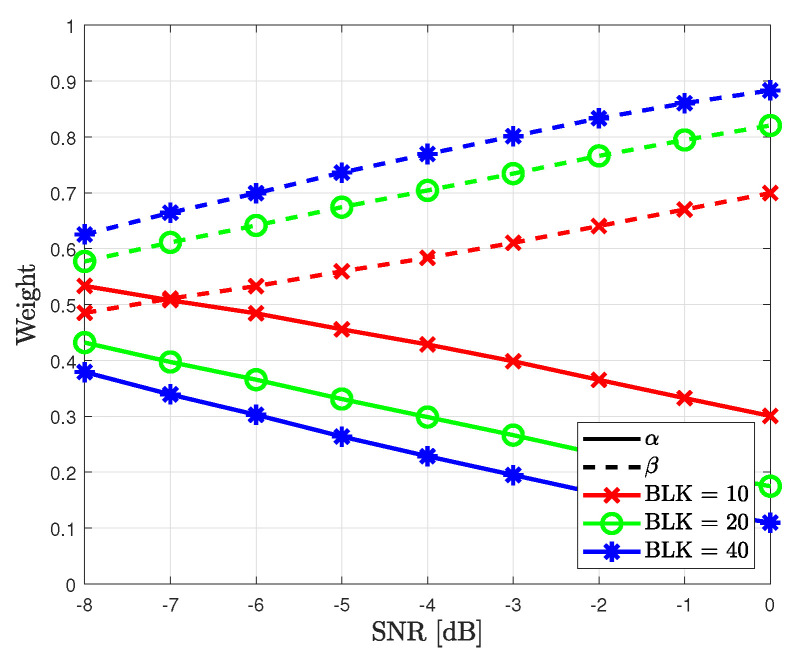
Combined weights for the proposed method based on SNR.

**Figure 5 sensors-23-05542-f005:**
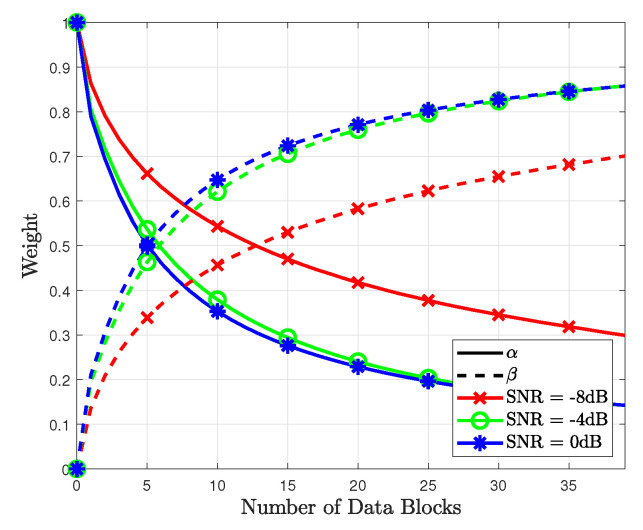
Combined weights for conventional RL based on number of data blocks.

**Figure 6 sensors-23-05542-f006:**
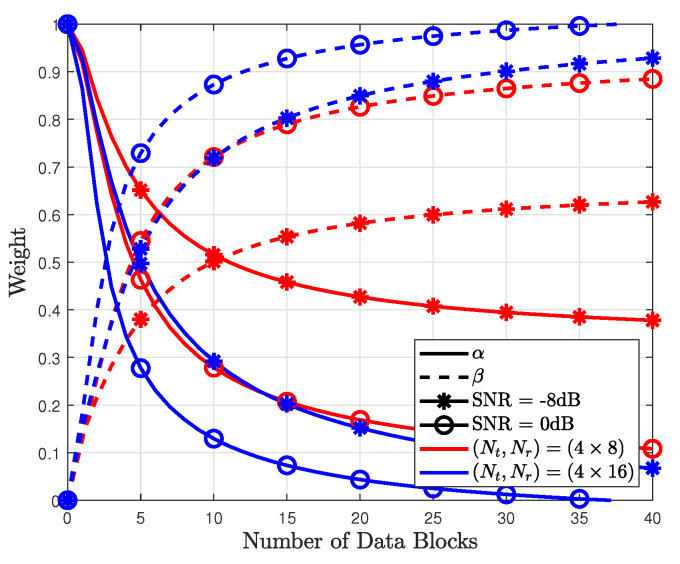
Combined weights for different number of antennas for $N_{r}=8$ and 16.

**Figure 7 sensors-23-05542-f007:**
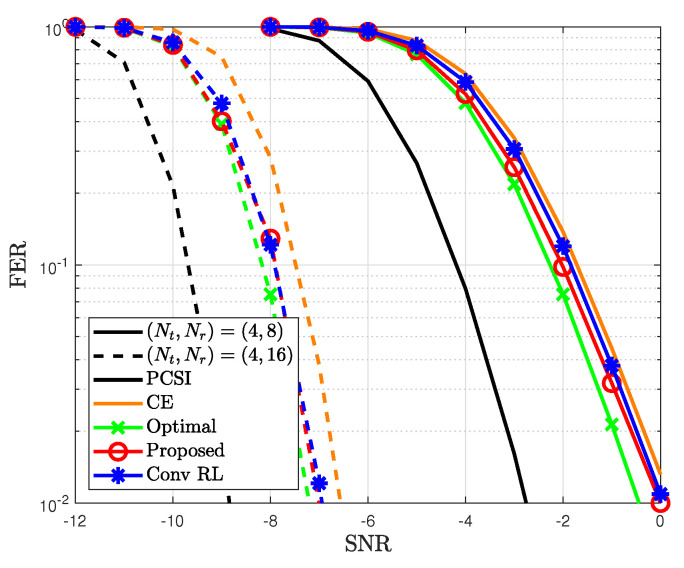
FER for the proposed method based on different detection methods.

**Figure 8 sensors-23-05542-f008:**
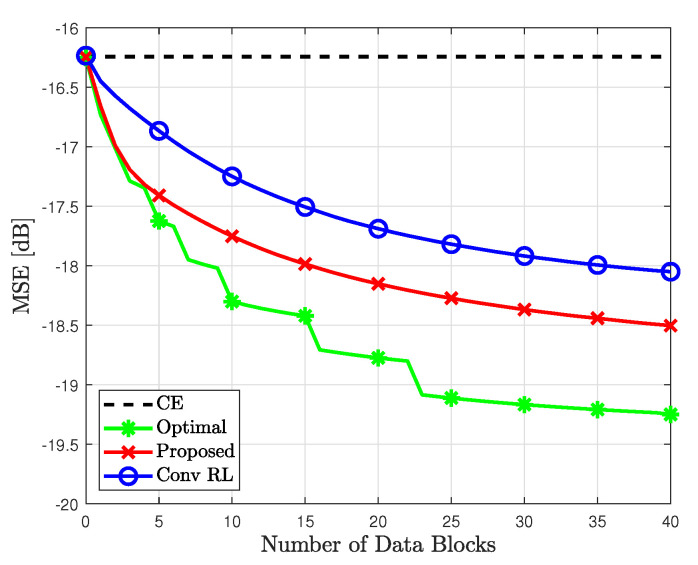
MSE for the proposed method based on different detection methods at SNR = 0 dB.

**Table 1 sensors-23-05542-t001:** Related works.

References	Topics	Key Features
[1,2,3,4,5,6]	Millimeter-wave system	Mmwave Channel model and implementation
[16,17,18,19,20,21]	Low-resolution ADCs	Optimal transceiver design
[22,23,24,25,26,27,28]	Channel estimation in one-bit ADCs	Optimal MMSE estimator design
[29,30,31,32,33,34,35]	Data detection in one-bit ADCs	Maximum likelihood probability calculation
[30,31,32,33]	Machine learning-based data detection	Machine learning-based algorithm

**Table 2 sensors-23-05542-t002:** Representative symbols in this study.

Symbols	Descriptions
H $\in R^{2N_{r}\times 2N_{t}}$	wireless channel matrix
s[n] $\in R^{2N_{t}}$	transmitted symbol at time *n*
$s_{k}$ $\in R^{2N_{t}}$	candidate transmitted symbol where k∈K
$\hat{k}\left[n\right]$ ∈K	detected symbol index at time *n*
n[n] $\in R^{2N_{r}}$	additive white Gaussian noise at time *n*
r[n] $\in R^{2N_{r}}$	received symbol at time *n*
q[n] $\in R^{2N_{r}}$	quantized signal at time *n*
$P\{q\left[n\right]|s\left[n\right]=s_{k}\}$	likelihood probability when symbol $s_{k}$ is transmitted
$p_{r,k}$	true likelihood probability $\Phi \left(\sqrt{\frac{2}{\sigma_{n}^{2}}}h_{r}^{H}s_{k}\right)$
$p_{r,k}^{pi}$	pilot-based likelihood probability $\Phi \left(\sqrt{\frac{2}{\sigma_{n}^{2}}}{\hat{h}}_{r}^{H}s_{k}\right)$
$p_{r,k}^{da}$	data-based likelihood probability $\frac{\sum_{m\leq n}{\tilde{q}}_{r}\left[m\right]1\left\{\hat{k}\left[m\right]=k\right\}}{\sum_{m\leq n}1\left\{\hat{k}\left[m\right]=k\right\}}$
${\hat{p}}_{r,k}$	estimated likelihood probability $\alpha p_{r,k}^{pi}+\beta p_{r,k}^{da}$

**Table 3 sensors-23-05542-t003:** Complexity comparison: the number of real multiplications.

Algorithm	Estimated Likelihood Probability Calculation		Total ${}^{1}$
	Optimal Policy Calculation	Likelihood Probability Update	
RL method [32]	$N_{r}N_{d}K+N_{r}N_{d}^{2}K+8N_{r}\left(K+1\right)\left(2K+1\right)N_{d}$	$4N_{r}KD$	$2.30\times 10^{9}$
Proposed method	-	$4N_{r}KD$	$3.27\times 10^{5}$

${}^{1}$ The number of real multiplications is counted when $N_{t}=4$, $N_{r}=8$, M=2, $N_{d}=128$, and D=40.

## Data Availability

Not applicable.

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
