# Peer review of "Improved Likelihood Probability in MIMO Systems Using One-Bit ADCs"

_sensors, 2023, doi:10.3390/s23125542_

Round 1

Reviewer 1 Report

Moderate editing of English language is needed.

Reviewer 2 Report

This paper presents a method for improving the likelihood probability in MIMO systems. Simulation results are provided to demonstrate the proposed method by enhancing the accuracy of the likelihood probability. Several detailed concerns are provided below.

1In Fig.8, why the optimal curve is not smooth? Please explain.

2For PER, the CE is higher than the proposed one, what is the possible reason?

3Some formulas for the system model are classical, they can be simplified.

4The authors could provide a detailed comparison between this work and those published related works, to further show the advantage and contribution.

Reviewer 3 Report

The paper proposes an improved likelihood probability method for multi-input multi-output (MIMO) systems that use one-bit ADCs. The paper is well written and the ideas of the paper are mostly well discussed. Nevertheless the paper has some flaws that must be addressed before the paper can be considered for publication.

Major concerns:

1) First, a "related works" section must be added to the article. The related works must be presented as well as their differences to the proposal shown in the submission. In my point of view the article needs a more solid result section. There is only one result regarding system performance with the proposed methodology (fig.7) and, even so, the proposed methodology presents an identical result to the one obtained by ConvRL (from the literature). The other results presented in the article only show the parameters and the MSE obtained by the optimization process. The following questions are not properly answered in the article: What is the advantage of the proposed method? In which aspects? What results support these advantages? What is the performance of the proposal in other modulation formats? These issues are crucial to be discussed in the article since the proposal reaches a result similar to that already found in the literature. 

Other comments:

1) "A one-bit operation is performed on the real and imaginary components of the received symbol r[n], which is expressed as"

Why  is one-bit operation given by the sing() function? Please elaborate on that on the manuscript and/or provide a reference.

2) What do you mena by "...where (20) is applied to (a)." ? Please elaborate the idea in the manuscript. 

3) How eq. 18 is derived from eq. 17 ? Please explain in the manuscript. Is the derivation your proposal or does it comes from the literature? 

4) "and coding schemes (see Fig. 2."

Missing closing parenthesis. 

5) "Saving all the parameters requires a large memory size, which increases the communication latency. "

It would be helpful to the paper if the authors could give simulations results for the latency

Reviewer 4 Report

In this manuscript, an improved likelihood probability in multi-input multi-output (MIMO) systems using one-bit analog-to-digital converters (ADCs) has been discussed. The MIMO systems using one-bit ADCs are known to exhibit from performance degradation because of inaccurate likelihood probabilities. As reviewer’s knowledge, this is a project in this research field. To overcome this degradation, the proposed method leverages the detected symbols to estimate the true likelihood probability by combining the initial likelihood probability; and an optimization problem is formulated to minimize the mean-squared error between the true and combined likelihood probabilities, as well as a good solution is derived using the least-squares method. This reviewer strongly recommends the manuscript accepted for publication. However, some minor corrections are needed.

1. There are more self-citations in the paper, some of them should be removed.

2. The conclusion should indicate what issues need further study.

Reviewer 5 Report

The manuscript needs some important revisions. Also, some sectors need a better justification for better comprehension. Please revise properly for the best possible result.

The candidate paper is relevant to a technique based on better likelihood probability. It is designed especially for MIMO systems with 1-bit ADCs.

This manuscript proposes a new technique relevant to a likelihood probability mechanism. The author mentions something important, but more examples should be given, probably with other modulation types. In some points, better writing is welcomed, and thus some grammatical and expressional issues should be attended to.

In addition to the previously mentioned, please apply these suggestions, and import the answers inside the manuscript:

  1. "Simulation results show that the proposed method outperforms conventional methods by improving the reliability of the likelihood probability". Please also give a quantitative result.
  2. "…in current wireless standards [1–6]." Why did you select these standards? Do they represent the whole category or else please use or add appropriate bibliography?
  3. You mention that "The transmitter has Nt antennas and the receiver has Nr antennas". What is a practical number of antennas? Please report it, otherwise please find through simulations an appropriate number not too large but still to exhibit significant efficiency as a large number of antennas do (such as N).
  4. Figure 7 findings are not explained adequately but rather chaotic. Please revise and be simple in your explanations.
  5. Again figure 8 needs a proper, and simple explanation. Please revise to show the impact of this figure.
  6. Conclusions should give a quantitative value of some important findings. Please revise.
  7. References should be enriched based on rules (e.g. which database has been used, with what keywords, and how many exclusions, etc.).

Moderate editing of English language

Round 2

Reviewer 2 Report

Thanks to the author for the revision. My concerns have been addressed.

Author Response

Thank you for taking time in reviewing my work.

Reviewer 3 Report

Thank you for updated versions of the manuscript.

I still think that this paper requires a most extent amount of simulation results to demonstrate the effectiveness of the proposed approach in different scenarios. However, the present form of the paper is acceptable. 

Author Response

(The authors gave the same response as above.)

Reviewer 5 Report

The author has made revisions. Please import the whole or partial the answer of suggested improvement (3). Also, please import the answer of the suggested improvement (7) in a proper manner (based on your criteria) for showing that you applied at least some categorization rules. You must also explain which database has been used, with what keywords, how many exclusions, etc, or else explain your method of finding the needed bibliography.

Minor editing of English language required
